# The Impact of Selected *COL1A1* and *COL1A2* Gene Polymorphisms on Bone Mineral Density and the Risk of Metabolic Diseases in Postmenopausal Women

**DOI:** 10.3390/ijms26114981

**Published:** 2025-05-22

**Authors:** Edyta Cichocka, Sylwia Barbara Górczyńska-Kosiorz, Paweł Niemiec, Wanda Trautsolt, Janusz Gumprecht

**Affiliations:** 1Department of Internal Medicine, Diabetology and Nephrology, Faculty of Medical Sciences in Zabrze, Medical University of Silesia, 40-055 Katowice, Poland; skosiorz@sum.edu.pl (S.B.G.-K.); wtrautsolt@sum.edu.pl (W.T.); jgumprecht@sum.edu.pl (J.G.); 2Department of Biochemistry and Medical Genetics, School of Health Sciences in Katowice, Medical University of Silesia, Medykow Street 18, 40-752 Katowice, Poland; pniemiec@sum.edu.pl

**Keywords:** *COL1A1*, *COL1A2*, gene polymorphism, bone mineral density, obesity, type 2 diabetes, postmenopausal women

## Abstract

Genetic variations in the *COL1A1* and *COL1A2* genes have been linked to bone mineral density (BMD) and metabolic disorders. This study analyzed the associations of *COL1A1* (rs1107946, rs1800012) and *COL1A2* (rs42524) polymorphisms with BMD, obesity, and type 2 diabetes (T2D) in 554 postmenopausal women. Dual-energy X-ray absorptiometry assessed BMD, and genotyping was performed alongside an evaluation of metabolic and lifestyle factors. The *COL1A1* rs1107946 AA genotype was associated with higher femoral neck BMD (*p* < 0.05), an over 10-fold increased obesity prevalence (*p* = 0.038), and a 3.5-fold higher T2D risk (*p* = 0.011)—a novel finding. The rs1800012 polymorphism showed age-dependent BMD effects: A allele carriers had lower femoral neck BMD in the 60–69 age group but higher total hip BMD in the 70–79 age group. Additionally, *COL1A2* rs42524 GG homozygotes had a significantly higher incidence of maternal fractures (*p* < 0.05). These results highlight *COL1A1* rs1107946 as a potential marker for both skeletal and metabolic risk, demonstrate the age-specific effects of rs1800012 on BMD, and identify rs42524 as a possible genetic indicator of familial fracture risk. These insights may inform personalized approaches to osteoporosis and metabolic disease prevention.

## 1. Introduction

Postmenopausal women are at a significantly increased risk of developing osteoporosis and metabolic disorders due to hormonal changes that affect bone remodeling and metabolism [1,2]. Type I collagen, the main structural protein of bone, is encoded by the *COL1A1* and *COL1A2* genes, making these genes critical determinants of bone mineral density (BMD) [3,4]. Among the genetic variants that influence bone health, the rs1107946 and rs1800012 polymorphisms of *COL1A1* and the rs42524 polymorphism of *COL1A2* have been extensively studied for their potential roles in bone fragility and fracture susceptibility [4,5,6].

The *COL1A1* gene is located on chromosome 17 (17q21.33). The gene variant rs1800012 [NC_000017.11:g.50200388C>A] is found in intron 1 and affects gene transcription [7,8]. Another variant, rs1107946 [NC_000017.11:g.50203629A>G], is located in the gene’s promoter region [9,10].

The rs1800012 polymorphism (Sp1 binding site) in *COL1A1* has been associated with an imbalance in the α1 to α2 collagen chain ratio, resulting in reduced bone strength and an increased risk of osteoporotic fractures [5,11]. Additionally, this polymorphism has been linked to intervertebral disk degeneration, ligament injuries, and an increased susceptibility to musculoskeletal conditions [12,13,14].

The rs1107946 variant has been investigated for its role in bone mass regulation, with evidence suggesting it may contribute to variations in BMD across different populations [3,15].

In the *COL1A2* gene, located on chromosome 7 (7q21.3), the rs42524 variant [NC_000007.14:g.94413927C>G], found in exon 28 [16,17], has been studied not only in the context of osteoporosis but also in relation to cardiovascular diseases and metabolic disorders [18]. This variant has been linked to alterations in collagen fibril formation, potentially influencing arterial stiffness, insulin sensitivity, and lipid metabolism [19].

Beyond their role in skeletal health, polymorphisms in the *COL1A1* and *COL1A2* genes have been implicated in the development of metabolic disorders such as obesity, type 2 diabetes mellitus (T2D), and cardiovascular disease [20]. Collagen abnormalities may influence extracellular matrix remodeling, adipose tissue function, and pancreatic β-cell activity, which are critical factors in metabolic regulation [20]. Recent studies suggest that alterations in collagen-related pathways may contribute to insulin resistance, dyslipidemia, and atherosclerosis, further linking these genetic variants to broader health concerns [21].

Given the complex interplay between genetic factors, bone health, and metabolic regulation, understanding the impact of *COL1A1* and *COL1A2* polymorphisms is essential for identifying at-risk individuals and developing targeted prevention strategies.

The aim of this study was to evaluate the association between rs1107946 and rs1800012 polymorphisms in *COL1A1*, as well as rs42524 in *COL1A2*, with BMD and the risk of metabolic disorders in postmenopausal women. By synthesizing current evidence, this article provides insights into the genetic determinants of bone and metabolic health, offering potential implications for personalized medicine and disease prevention.

## 2. Results

### 2.1. General Characteristics of the Study Group

All examined women were postmenopausal. Osteoporosis (T-score ≤ −2.5) was diagnosed in over 9% of the women based on femoral neck BMD T-scores and in 4% based on total hip BMD T-scores. Osteopenia (T-score ≤ −1, >−2.5) was identified in 50.09% of the women according to femoral neck BMD T-scores and in 29.80% based on total hip BMD T-scores. Additional clinical data are presented in Table 1.

### 2.2. COL1A1 and COL1A2 Gene—Polymorphisms

The genotype and allele frequencies of the analyzed *COL1A1* and *COL1A2* gene polymorphisms are presented in Table 2. The genotype distribution was consistent with a Hardy–Weinberg equilibrium (*p* > 0.050).

The *COL1A1* gene polymorphisms do not form a haplotype, as indicated by a low R^2^ value (0.037) despite a high D’ value (1.00). Similar patterns are observed in the CEU population (Utah residents of Northern and Western European ancestry, GRCh38).

### 2.3. COL1A1 and COL1A2 Gene Polymorphisms and BMD

The association between *COL1A1* and *COL1A2* gene polymorphisms and BMD was assessed based on densitometric measurements of the femoral neck (FN) and total hip (TH).

In the additive model (genotype-based comparison), no statistically significant differences in BMD were observed. However, in the recessive/dominant model (carrier-based analysis), individuals homozygous for the AA genotype of the *COL1A1* rs1107946 polymorphism exhibited higher BMD values across all measured parameters compared to carriers of the C allele. Statistically significant differences were found specifically for FN BMD and FN T-score (Table 3, Figure 1).

A detailed distribution of quantitative BMD parameters for AA homozygotes and C allele carriers (*COL1A1* rs1107946 polymorphism) is presented in Figure 1.

Due to the decline in BMD with age, further analysis was conducted to evaluate BMD parameters across different age groups based on the genotype variants of the studied polymorphisms. For the *COL1A1* rs1800012 polymorphism, statistically significant differences in DXA parameters were observed between genotypic variants. These differences were identified in the recessive/dominant model (AA/AC vs. CC). In the 60–69 age group, A allele carriers had lower femoral neck BMD (BMD FN) compared to CC homozygotes (Figure 2). Interestingly, in the 70–79 age group, A allele carriers exhibited more favorable total hip BMD (BMD TH) and TH T-score values, suggesting that the observed effect may be age-dependent.

No significant associations between other polymorphisms and BMD were found across different age groups.

### 2.4. Family History of Fractures

A significant association between the *COL1A2* rs42524 polymorphism and a family history of fractures was observed. GG homozygotes had a higher prevalence of positive fracture history compared to C allele carriers, with statistically significant differences identified in the recessive/dominant model (Table 4).

When the positive family history of fractures was maternal, the observed differences were more pronounced and reached statistical significance in both the additive and recessive/dominant models (Table 4). However, no statistically significant differences in rs42524 genotype distribution were found between women with a positive or negative paternal history of fractures (Table 4).

### 2.5. COL1A1 and COL1A2 Gene Polymorphisms and Their Association with BMI, Overweight, and Obesity

The potential differences in BMI values between *COL1A1* and *COL1A2* gene polymorphism variants were analyzed. Individuals homozygous for the AA genotype of the *COL1A1* rs1107946 polymorphism had a significantly higher BMI compared to C allele carriers (median ± QD: 36.21 ± 5.36 vs. 30.78 ± 7.86, respectively; *p* = 0.020). No statistically significant differences in BMI were observed for the other polymorphisms studied.

Next, the genotype distribution of the rs1107946 polymorphism was examined across different BMI categories: normal weight (BMI < 25), overweight (BMI ≥ 25 < 30), and obesity (BMI ≥ 30) (Figure 3). The frequency of the AA genotype was over 10 times higher in women with obesity compared to those in other BMI categories (Figure 3A) (*p* = 0.038). Furthermore, an analysis of BMI category distribution across rs1107946 genotypes (Figure 3B) revealed that 84.62% of AA homozygotes in the study group were classified as obese, whereas the prevalence of obesity among individuals with other genotypes was 53.29% (AC) and 54.75% (CC) (*p* = 0.038 in the additive model, *p* = 0.068 in the recessive/dominant model: AA vs. AC/CC).

### 2.6. COL1A1 and COL1A2 Gene Polymorphisms and Diabetes

The distribution of *COL1A1* and *COL1A2* polymorphisms was analyzed in relation to the presence of T1D and T2D in the study group. Among the studied polymorphisms, only *COL1A1* rs1107946 showed statistically significant differences in genotype frequencies between women with T2D and those without the disease (Figure 4A). The frequency of the AA genotype was more than 3.5 times higher in women with T2D compared to non-diabetic participants (6.17% vs. 1.69%, OR = 3.82, 95% CI: 1.22–12.03, *p* = 0.011).

An analysis of T2D distribution across genotypes (Figure 4B) revealed that 38.46% of AA homozygotes in the study group had T2D, whereas the prevalence was 9.87% in AC heterozygotes and 15.68% in CC homozygotes (*p* = 0.011 in the additive model, *p* = 0.039 in the recessive/dominant model: AA vs. AC/CC).

None of the analyzed polymorphisms were associated with T1D or diabetes in general when considering both primary types of the disease together.

### 2.7. COL1A1 and COL1A2 Gene Polymorphisms and Confounding Factors

The potential differences in the frequency of factors that could influence BMD results were assessed (Table 5). No statistically significant differences were found in the distribution of these factors across the genotypes of the analyzed polymorphisms (*p* > 0.050).

## 3. Discussion

Our study provides novel insights into the genetic associations of *COL1A1* and *COL1A2* polymorphisms with BMD and metabolic disorders. While previous research has explored the role of collagen-related genes in skeletal fragility and metabolic syndromes, we present pioneering findings that expand the current understanding.

### 3.1. COL1A1 rs1107946 and BMD

The *COL1A1* rs1107946 AA genotype was linked to increased BMD, particularly in the femoral neck region. This finding aligns with previous studies suggesting that a higher body weight is correlated with increased bone density due to mechanical loading effects [22]. However, this is the first time that a direct genetic association between this polymorphism and BMD is reported.

Our findings are consistent with a study conducted among Slovak postmenopausal women which evaluated the association between the COL1A1 rs1107946 polymorphism, BMD, and fracture risk. That study demonstrated that individuals carrying the A allele had significantly higher BMD values, particularly at the femoral neck, and a lower incidence of fractures compared to those with the GG genotype. These results support our observation that rs1107946 may play a protective role in bone health, potentially through its impact on collagen structure or bone remodeling processes [23].

Taken together, these findings suggest that rs1107946 is not only relevant in population-specific contexts but may have broader implications as a genetic marker for increased bone strength and reduced fracture susceptibility. Further studies in diverse cohorts are needed to validate its utility in clinical risk stratification.

### 3.2. COL1A1 rs1800012 and Age-Dependent BMD Effects

Our study also confirms an age-dependent effect of the *COL1A1* rs1800012 polymorphism on BMD. A allele carriers exhibited lower femoral neck BMD in the 60–69 age group but had higher total hip BMD in the 70–79 age group. These findings suggest that genetic influences on bone density may not be static but rather evolve with age. While previous research has identified *COL1A1* rs1800012 as a risk factor for osteoporosis [5], its age-specific effects have not been well documented. One possible explanation is that hormonal changes, lifestyle factors, or other age-related biological shifts may modify the impact of this polymorphism on bone metabolism over time. Another study that examined the association between COL1A1 polymorphisms and BMD in a Polish cohort was performed in 2016 [24]. The study found no significant association between the analyzed polymorphisms and BMD. This discrepancy may arise from differences in sample size, population genetics, or environmental factors influencing bone health. Our results are also supported by the Dubbo Osteoporosis Epidemiology Study, which evaluated COL1A1 Sp1-binding-site polymorphisms (rs1800012) in 809 postmenopausal women over a 30-year period. That study found that homozygous carriers of the minor allele (TT) had the greatest rate of bone loss and were at significantly increased risk for fragility fractures, particularly hip fractures, with a nearly fourfold increased risk compared to GG or GT genotypes. Notably, this increased fracture risk was independent of bone loss, suggesting that COL1A1 variants may affect bone quality or structural integrity in ways not fully captured by BMD alone. The Dubbo study reinforces our findings and underscores the clinical relevance of rs 1,800,012 in assessing fracture risk. It also suggests that the fracture risk associated with this polymorphism might not be entirely mediated by changes in BMD, pointing toward potential impacts on collagen quality or bone microarchitecture. This highlights the need for integrating genetic profiling in future osteoporosis risk models, especially when evaluating older adults with borderline BMD values or unexplained fragility fractures [24]. Additional longitudinal studies are needed to explore this dynamic relationship.

### 3.3. COL1A2 rs42524 and Maternal Fracture History

One of the most significant and novel findings of this study is the association between the *COL1A2* rs42524 polymorphism and a maternal history of fractures. This is the first study that has confirmed this relationship. GG homozygotes in the *COL1A2* rs42524 demonstrated a significant relationship with a family history of fractures, particularly maternal fractures. A study conducted among postmenopausal Chinese women investigated polymorphisms in the *COL1A1* and *COL1A2* genes to assess their impact on fracture onset. The research concluded that these genetic variations did not significantly affect fracture risk in the studied population [25]. This contrasts with our findings, particularly regarding the *COL1A2* rs42524 polymorphism’s association with maternal fracture history, suggesting potential ethnic or population-specific genetic influences. Given that type I collagen plays a crucial role in bone strength, this association warrants further investigation to determine whether *COL1A2* variants contribute to inherited osteoporosis risk. Future studies should explore whether this genetic variant affects collagen structure, bone turnover, or fracture susceptibility in different populations.

### 3.4. COL1A1 rs1107946 and Obesity

We found a strong relationship between the *COL1A1* rs1107946 AA genotype and obesity. Our results show that the AA genotype is over 10 times more common in obese individuals compared to other BMI categories, a finding that has not been previously reported. While *COL1A1* is primarily associated with bone formation and connective tissue integrity [26], emerging evidence suggests a potential role in adipogenesis and metabolic regulation. Some studies have linked *COL1A1* variants to altered extracellular matrix composition, which could influence fat accumulation and metabolic function [27,28]. However, our findings indicate a need for further research into how *COL1A1* polymorphisms may regulate body weight and adipose tissue distribution.

### 3.5. COL1A1 rs1107946 and T2D

Our study also identifies a novel association between the *COL1A1* rs1107946 AA genotype and an increased risk of type 2 diabetes. The AA genotype was 3.5 times more prevalent in women with T2D compared to non-diabetic participants. To date, no studies have explored this relationship, making this an important finding that could contribute to the understanding of genetic risk factors for metabolic diseases. Given the strong link between obesity and T2D, it is possible that the *COL1A1* rs1107946 variant influences metabolic pathways that predispose individuals to both conditions. Further research is needed to investigate whether this polymorphism affects insulin resistance, glucose metabolism, or inflammatory pathways, which are key contributors to diabetes pathophysiology.

## 4. Materials and Methods

### 4.1. Study Group

This study was conducted as part of the RAC-OST-POL project, a retrospective epidemiological cohort study. Approval was granted by the Bioethics Committee of the Medical University of Silesia, Katowice, Poland (No. KNW/0022/KB1/9/I/10 date 2 May 2010), and the research was carried out in compliance with the STROBE guidelines [29].

Participants in the research project were randomly selected from a population of over 17,500 postmenopausal women aged 55 and older living in Racibórz, a town in the Upper Silesia region of southern Poland (the average age of menopause in Poland is 49). Invitations were mailed via regular post to 1750 women—representing 10% of this demographic group. A total of 625 women responded to the invitation. All of them belonged to the Caucasian population. For the final analysis, only those with complete genotyping data were included, resulting in a final sample of 554 women. The included women underwent bone mineral density assessments. The inclusion criteria were informed consent to participate in this study and postmenopausal status, defined as at least one year since the last menstrual period. The exclusion criteria were a lack of an invitation to participate and an absence of informed consent. Genetic analysis was performed to determine the allele and genotype frequencies of the rs1107946 and rs1800012 polymorphisms in the *COL1A1* gene and rs42524 in the *COL1A2* gene. Additionally, anthropometric measurements were recorded, and data on calcium and vitamin D3 supplementation, as well as on the use of anti-osteoporotic medications, were collected.

Within this subgroup, detailed information was collected regarding comorbidities (arterial hypertension, diabetes, steroid therapy, thyroid disorders, rheumatoid arthritis, asthma, chronic kidney disease, depression, Alzheimer’s disease), fracture occurrence, family history of fractures, smoking habits, and alcohol consumption. Only coexisting medical conditions that may affect BMD values were accounted for in the analysis.

Among the total study population, 81 women were diagnosed with type 2 diabetes (T2D) and 19 women with type 1 diabetes (T1D).

### 4.2. Bone Mineral Density Measurement

Bone mineral density (BMD) measurements were performed using a Lunar DPX device (GE, Madison, WI, USA). The assessments focused on the non-dominant femoral neck (FN) and hip joint (HJ) in female participants. BMD values were expressed in standardized units (g/cm^2^) and calculated using T-scores based on reference data from the National Health and Nutrition Examination Survey (NHANES) for white women aged 20 to 29 years. Osteoporosis was diagnosed according to the WHO criteria [30].

All measurements were conducted by a single experienced operator to ensure consistency. The coefficient of variation (CV%), determined from 50 measurements, was 1.6% for the femoral neck and 0.82% for the hip joint.

### 4.3. Genetic Analyses

Venous blood samples were collected from all participants. DNA was isolated from blood samples frozen at −20 °C and collected in EDTA tubes. The MasterPure DNA Kit (Epicenter Technologies, Madison, WI, USA) was used for genomic DNA extraction. Genotyping was performed based on single-nucleotide polymorphism (SNP) analysis of the *COL1A1* and *COL1A2* genes for rs1800012, rs1107946, and rs42524 using TaqMan Predesigned SNP Genotyping Assay Kits (Thermo Fisher Scientific, Carlsbad, CA, USA). Amplification was carried out using the 7300 Real-Time PCR System (Thermo Fisher Scientific, Carlsbad, CA, USA). Genotyping accuracy was verified by re-genotyping 10–15% of the samples, achieving 100% repeatability.

### 4.4. Statistical Analysis

Statistical analysis was performed using Statistica 13.1 (TIBCO Software Inc., Santa Clara, CA, USA). The distribution of the data was evaluated using the Shapiro–Wilk test. Quantitative data that did not have a normal distribution were presented as medians with quartile deviation (QD). The Mann–Whitney U test was used to represent dichotomous grouping variables. The Kruskal–Wallis test, along with post hoc analysis, was used to compare three or more groups for a given quantitative variable. The relationship between the study variables was assessed using Spearman’s rank correlation coefficient. The Hardy–Weinberg equilibrium was assessed using the x2 test and comparisons of genotype and allele frequencies between groups, differentiated by qualitative variables. For subgroups with fewer than ten patients, Fisher’s correction was used.

The medians of quantitative variables, such as DXA 141 test scores and obesity surrogates, were compared in an additive inheritance model (between individual genotypes) and a recessive/dominant model (between the carriers of individual alleles). Qualitative variables (a history of fractures, the diagnosis of osteoporosis according to WHO (T-score ≤ −2.5), the use of anti-osteoporotic therapy, and the diagnosis of obesity, diabetes, and others, were assessed for significant statistical differences in genotype frequencies among the classes of these variables. A *p*-value of less than 0.050 was considered statistically significant. Sample size and power analysis were also computed using Statistica 13.1 software. The power of all statistically significant tests in the present study was greater than 80%, with a 95% two-sided confidence level. For multiple comparisons, *p*-values were corrected using the Bonferroni correction.

## 5. Conclusions

In summary, this study identifies novel genetic associations between *COL1A1* and *COL1A2* polymorphisms and key health outcomes in postmenopausal women. The pioneering findings related to *COL1A2* rs42524 and maternal fracture history, *COL1A1* rs1107946 and obesity/T2D, and the age-dependent effects of *COL1A1* rs1800012 on BMD underscore the importance of genetic factors in bone and metabolic health.

A major strength of this study is that it was conducted on a randomly selected group of ethnically and genetically homogeneous women from the general population, all residing in the same geographical area. This enhances the reliability of the results obtained. These findings require confirmation in other populations and ethnic groups. A limitation of this study is the lack of assessment of the biochemical profile (i.e., HbA1c, lipid profile), as well as the absence of data regarding diabetes treatment regimens and metabolic control, which would have allowed for a more comprehensive analysis in the context of metabolic diseases. Nevertheless, our findings offer potential implications for personalized approaches to osteoporosis and metabolic disorder prevention. Future research should further explore these associations in larger cohorts and investigate potential underlying mechanisms to enhance our understanding of genetic contributions to postmenopausal health.

## Figures and Tables

**Figure 1 ijms-26-04981-f001:**
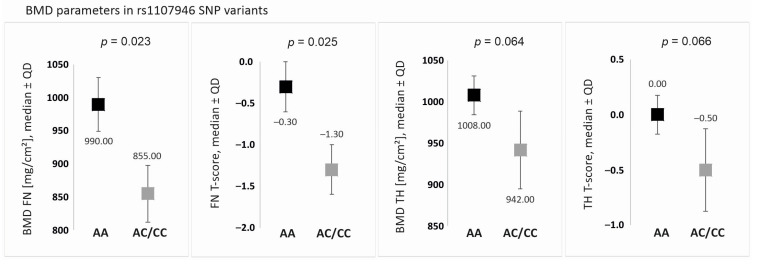
Bone mineral density (BMD) parameters for the genotype variants of the *COL1A1* rs1107946 SNP. Data available for 547 out of 554 subjects due to insufficient DNA quality. Legend: BMD, bone mineral density; FN, femoral neck; TH, total hip; QD, quartile deviation.

**Figure 2 ijms-26-04981-f002:**
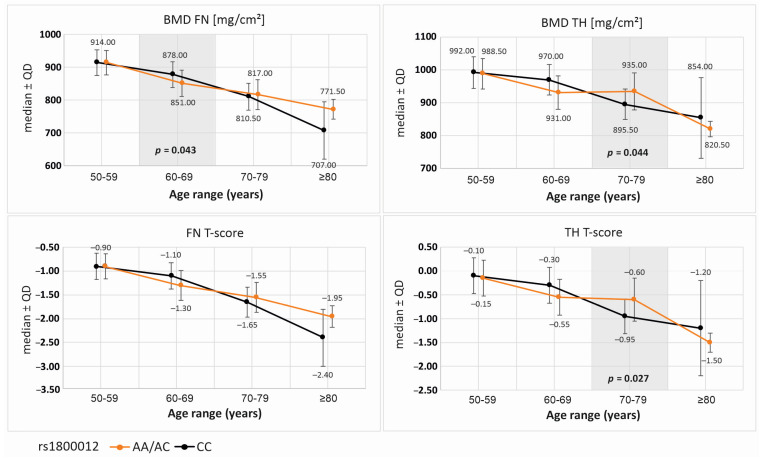
Bone mineral density (BMD) parameters for the genotype variants of the *COL1A1* rs1800012 SNP in respective age ranges. Data available for 547 subjects. Legend: BMD, bone mineral density; FN, femoral neck; TH, total hip; QD, quartile deviation.

**Figure 3 ijms-26-04981-f003:**
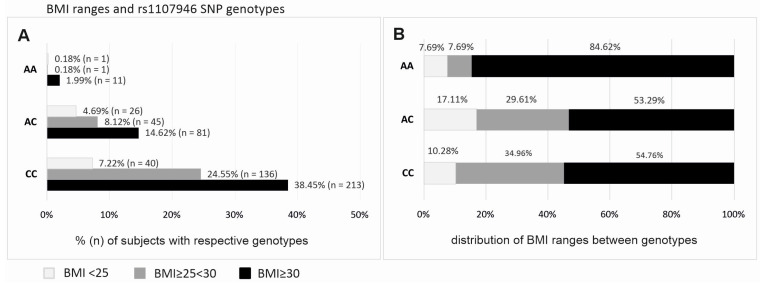
Percentage and number of subjects with the respective genotypes of rs1107946 of *COL1A1* gene in respect to BMI (**A**) and the distribution of BMI ranges between genotypes (**B**).

**Figure 4 ijms-26-04981-f004:**
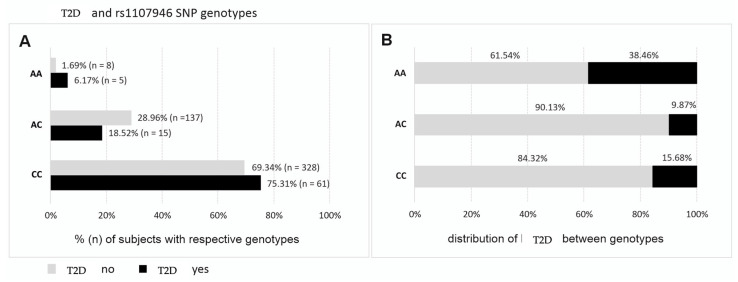
Percentage and number of subjects with the respective genotypes of rs1107946 of *COL1A1* gene in women with T2D (**A**) and the distribution of T2D between genotypes (**B**).

**Table 1 ijms-26-04981-t001:** Basic demographic and clinical characteristics of the study group.

Characteristics			
**General**	number of subjects, n (%)	554	(100.00)
	age [years], median ± QD	66.02	6.31
	age range [years], n (%)		
	- 50–59	143	(25.81)
	- 60–69	215	(38.81)
	- 70–79	166	(29.96)
	- 80 and more	30	(5.42)
	years after menopause [years], median ± QD	16.92	6.56
	BMI [kg/m^2^], median ± QD	30.85	3.93
	obesity [BMI ≥ 30], n (%)	306	(55.23)
	cigarette smokers, n (%)	62	(11.19)
	alcohol consumption [≥3 units/day], n (%)	4	(0.72)
**BMD parameters ***	BMD FN [mg/cm^2^], median ± QD	857.00	88.50
	BMD FN T-score, median ± QD	−1.30	1.20
	- normal BMD [T-score > −1], n (%)	222	(40.59)
	- osteopenia [T-score ≤ −1, >−2.5], n (%)	274	(50.09)
	- osteoporosis [T-score ≤ −2.5], n (%)	51	(9.32)
	BMD TH [mg/cm^2^], median ± QD	945.00	94.00
	BMD TH T-score, median ± QD	−0.50	1.50
	- normal BMD [T-score > −1], n (%)	362	(66.18)
	- osteopenia [T-score ≤ −1, >−2.5], n (%)	163	(29.80)
	- osteoporosis [T-score ≤ −2.5], n (%)	22	(4.02)
**Comorbidities**	T1D, n (%)	19	(3.43)
	T2D, n (%)	81	(14.62)
	glucocoticosteroid therapy, n (%)	24	(4.33)
	rheumatoid arthritis, n (%)	38	(6.86)
	thyroid gland diseases, n (%)	5	(0.90)
	chronic kidney disease, n (%)	6	(1.08)

Legend: BMD, bone mineral density; BMI, body mass index; FN, femoral neck; T1D, type 1 diabetes mellitus; T2D, type 2 diabetes mellitus; TH, total hip; QD, quartile deviation; *—data available for 547 subjects.

**Table 2 ijms-26-04981-t002:** Frequency of genotypes and alleles of analyzed SNPs of *COL1A1* and *COL1A2* genes.

Gene	SNP	Chromosome: Coordinate (GRCh38)	Genotypes	n (%)	Alleles	n (%)	*p* Value HWE Test
*COL1A1*	rs1107946	chr17:50203629	AA	13 (2.34)	A	178 (16.06)	0.931
			AC	152 (27.44)	C	930 (83.94)	
			CC	389 (70.22)			
*COL1A1*	rs1800012	chr17:50200388	AA	17 (3.07)	A	202 (18.23)	0.882
			AC	168 (30.32)	C	906 (81.77)	
			CC	369 (66.61)			
*COL1A2*	rs42524	chr7:94413927	CC	37 (6.68)	C	301 (27.17)	0.681
			CG	227 (40.97)	G	807 (72.83)	
			GG	290 (52.35)			

Legend: GRCh38, Genome Reference Consortium Human Build 38 Organism: *Homo sapiens*; HWE, Hardy–Weinberg equilibrium; SNP, single-nucleotide polymorphism.

**Table 3 ijms-26-04981-t003:** Bone mineral density (BMD) parameters in regard to the genotypes of analyzed *COL1A1* and *COL1A2* gene single-nucleotide polymorphisms (SNPs). Data available for 547 out of 554 subjects due to insufficient DNA quality.

BMD Parameter	SNP	Genotypes, Median ± QD			*p* Value for Models	
					Additive	Recessive/Dominant
	rs1107946	AA	AC	CC		AA vs. AC/CC
FN [mg/cm^2^]		990.00 ± 81.00	855.00 ± 87.50	855.00 ± 86.00	0.074	0.023
FN T-score		−0.30 ± 0.60	−1.30 ± 0.60	−1.30 ± 0.65	0.075	0.025
TH [mg/cm^2^]		1008.00 ± 47.00	945.00 ± 87.00	941.00 ± 100.50	0.174	0.064
TH T-score		0.00 ± 0.35	−0.50 ± 0.70	−0.50 ± 0.80	0.183	0.066
	rs1800012	AA	AC	CC		AA vs. AC/CC
FN [mg/cm^2^]		825.50 ± 61.00	855.00 ± 85.50	862.00 ± 0.89	0.596	0.328
FN T-score		−1.50 ± 0.43	−1.30 ± 0.60	−1.30 ± 0.60	0.586	0.305
TH [mg/cm^2^]		937.00 ± 104.50	935.00 ± 95.50	945.00 ± 88.50	0.907	0.694
TH T-score		−0.55 ± 0.83	−0.55 ± 0.75	−0.50 ± 0.70	0.835	0.717
	rs42524	CC	CG	GG		GG vs. CG/CC
FN [mg/cm^2^]		857.00 ± 80.50	859.00 ± 106.25	856.00 ± 81.50	0.910	0.676
FN T-score		−1.30 ± 0.60	−1.30 ± 0.78	−1.30 ± 0.55	0.886	0.625
TH [mg/cm^2^]		959.00 ± 93.50	947.50 ± 109.25	942.00 ± 90.50	0.980	0.944
TH T-score		−0.40 ± 0.75	−0.50 ± 0.88	−0.50 ± 0.75	0.946	0.750

Legend: BMD, bone mineral density; FN, femoral neck; SNP, single-nucleotide polymorphism; TH, total hip; QD, quartile deviation.

**Table 4 ijms-26-04981-t004:** Family history of fractures in respect to *COL1A2* gene rs42524 variants.

Positive Family History of Fractures		Genotypes, n (%)			*p* Value for Models:	
					Additive	Recessive/Dominant
		CC	CG	GG		GG vs. CC/CG
on both parents’ sides	Yes	7 (18.92)	49 (21.59)	86 (29.66)	0.071	0.023 ^1^
	No	30 (81.08)	178 (78.41)	204 (70.34)		
on the mother’s side	Yes	6 (16.00)	38 (17.00)	74 (26.00)	0.040	0.011 ^2^
	No	31 (84.00)	189 (83.00)	216 (74.00)		
on the father’s side	Yes	2 (5.41)	14 (6.55)	19 (6.17)	0.957	0.812
	No	35 (94.59)	213 (93.45)	271 (93.83)		

Legend: ^1^ OR = 1.57 (95% CI: 1.06–2.31), *p* = 0.023. ^2^ OR = 1.71 (95% CI: 1.29–2.60), *p* = 0.011.

**Table 5 ijms-26-04981-t005:** Distribution of confounding factors between genotypes of analyzed *COL1A1* and *COL1A2* gene single-nucleotide polymorphisms (SNPs).

Potential Confounder		SNP	Genotypes, n (%)			*p* Value
		rs1107946	AA	AC	CC	
rheumatoid arthritis	Yes		0 (0.00)	10 (1.81)	28 (5.05)	0.593
	No		13 (2.35)	142 (25.63)	361 (65.16)	
anti-osteoporotic therapy	Yes		3 (0.54)	27 (4.87)	57 (10.29)	0.510
	No		10 (1.81)	125 (22.56)	332 (59.93)	
Ca^2+^ supplementation	Yes		3 (0.54)	26 (4.69)	55 (9.93)	0.498
	No		10 (1.81)	126 (22.74)	334 (60.29)	
D vit. supplementation	Yes		3 (0.54)	24 (4.33)	47 (8.48)	0.304
	No		10 (1.81)	128 (23.10)	342 (61.73)	
		rs1800012	AA	AC	CC	
rheumatoid arthritis	Yes		2 (0.36)	6 (1.08)	30 (5.42)	0.099
	No		14 (2.53)	163 (29.42)	339 (61.19)	
anti-osteoporotic therapy	Yes		1 (0.18)	28 (5.05)	58 (10.47)	0.556
	No		15 (2.71)	141 (25.45)	311 (56.14)	
Ca^2+^ supplementation	Yes		1 (0.18)	26 (4.69)	57 (10.29)	0.601
	No		15 (2.71)	143 (25.81)	312 (56.32)	
D vit. supplementation	Yes		1 (0.18)	24 (4.33)	49 (8.84)	0.669
	No		15 (2.71)	145 (26.17)	320 (57.76)	
		rs42524	CC	CG	GG	
rheumatoid arthritis	Yes		2 (0.36)	17 (3.07)	19 (3.43)	0.858
	No		35 (6.32)	210 (37.91)	271 (48.92)	
anti-osteoporotic therapy	Yes		5 (0.90)	36 (6.50)	46 (8.30)	0.931
	No		32 (5.78)	191 (34.48)	244 (44.04)	
Ca^2+^ supplementation	Yes		5 (0.90)	34 (6.14)	45 (8.12)	0.945
	No		32 (5.78)	193 (34.84)	245 (44.22)	
D vit. supplementation	Yes		6 (1.08)	29 (5.23)	39 (7.04)	0.848
	No		31 (5.60)	198 (35.74)	251 (45.31)	

Legend: SNP, single-nucleotide polymorphism; vit., vitamin.

## Data Availability

Data are contained within the article.

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
