# Peer review of "The Impact of Selected COL1A1 and COL1A2 Gene Polymorphisms on Bone Mineral Density and the Risk of Metabolic Diseases in Postmenopausal Women"

_ijms, 2025, doi:10.3390/ijms26114981_

Round 1
Reviewer 1 Report
Comments and Suggestions for Authors
The present research article provides new and interesting insights into the association of selected polymorphisms of candidate genes COL1A1 and COL1A2 with bone mineral density and risk of metabolic disease in postmenopausal women.
Comprehensively I evaluate the article positively, I have some suggestions for its improvement:
- Add material and methods to include inclusion and exclusion criteria for women in the study.
- Add date of ethics committee approval.
- Comorbidities - Was there only 1 comorbidity in women or were there 2 or more? How many cases were there? Could they have affected the BMD value?
- What nationality were the group of postmenopausal women studied?
- Line 127, missing dot at the end of the sentence.
- Table 1 - BMD parameters - standardize the writing of T-scores with a capital T.
- In Table 2, correct the numbers for SNPs and genes in 1 line.
- Table 3, Figure 1 you state: Data available for 547 subjects. Why not 554 subjects? Please explain.
- Figure 3 - title stylization and typing correction.
- Table 4 - in the legend is missing a bracket.
- In subchapters 3.5 and 3.6, I would suggest that the correct frequencies of the individual genotypes are given in the text - do not round the numbers.
- I suggest expanding the discussion. Compare the results with several authors who have addressed this issue.
Author Response
Dear Reviewer,
I would like to thank you for the thorough evaluation and constructive feedback, which significantly helped to improve the quality and clarity of our manuscript. Please find below our detailed responses to each point raised. All suggested corrections and improvements have been incorporated into the revised version of the manuscript.
- Inclusion and exclusion criteria
I have added to the subsection Materials and Methods outlining the inclusion and exclusion criteria used in participant selection.
- Date of ethics committee approval
The date of approval by the local Ethics Committee has been included in the Materials and Method ssection.
- Comorbidities and their influence on BMD
I have expanded the description of coexisting conditions in the manuscript. The number of comorbidities and their potential impact on BMD per subject has been clarified.
- Nationality of participants
I clarified that all participants were of Polish nationality in the Materials and Methods section.
- Line 127 – punctuation
The missing period at the end of the sentence in Line 127 has been added.
- Table 1 – T-score formatting
All instances of “T-score” in Table 1 have been standardized using a capital “T.”
- Table 2 – SNP and gene alignment
The layout and formatting of Table 2 have been corrected to ensure consistent alignment of SNPs and gene names in each row.
- Table 3 and Figure 1 – discrepancy in subject number
I have added an explanatory note in the legends of Table 3 and Figure 1, stating that genotyping data were available for 547 out of 554 participants due to sample quality issues.
- Figure 3 – title formatting and typing
The title of Figure 3 has been revised for correct stylization and any typographical errors have been corrected.
- Table 4 – missing bracket
The missing bracket in the legend of Table 4 has been added.
- Subchapters 3.5 and 3.6 – genotype frequencies
I have updated the text in subchapters 3.5 and 3.6 to include the exact genotype frequencies without rounding.
- Expansion of the discussion
The Discussion section has been expanded to include comparisons with findings from multiple relevant studies. These additional references provide a broader context for our results and highlight how our findings relate to existing literature.
We appreciate the reviewer’s suggestion to expand the Discussion section by incorporating comparisons with findings from multiple relevant studies. We fully agree that placing our results in the context of existing literature enhances the scientific rigor and relevance of our work.
In response, we have revised the Discussion section to include comparisons with two additional peer-reviewed studies:
The Dubbo Osteoporosis Epidemiology Study, which examined the association of the COL1A1 rs1800012 polymorphism with bone loss and fracture risk over a 30-year period in postmenopausal women.
A Slovak study evaluating the COL1A1 rs1107946 polymorphism in relation to BMD and fracture risk in a cohort of postmenopausal women.
These studies provide valuable context and reinforce the clinical significance of our findings, particularly in terms of fracture risk and age-related genetic effects.
However, we would also like to emphasize that the overall body of literature directly addressing COL1A1 and COL1A2 polymorphisms in relation to bone mineral density and metabolic disorders especially in well-defined postmenopausal populations—remains limited.
Nevertheless, we believe the revised Discussion now presents a well-rounded synthesis of the most relevant available evidence, highlighting both the consistencies and the novel aspects of our findings. We hope these revisions meet the reviewer’s expectations and contribute meaningfully to the manuscript’s quality and impact.
Sincerely,
Edyta Cichocka
On behalf of all co-authors
Reviewer 2 Report
Comments and Suggestions for Authors
The aim of this study was to evaluate the association between rs1107946 and rs1800012 polymorphisms in COL1A1, as well as rs42524 in COL1A2, with BMD and the risk of metabolic disorders in postmenopausal women.
The introduction is appropriate to the aims of the study by indicating the relationship between these polymorphisms, metabolic bone disease and alterations in energy metabolism.
The methods are clear and easy to repeat by another research group. There is no indication of whether patients have been treated with drugs or the statistical power of the study.
The results are clearly expressed and easy to follow. The authors should indicate if patients with fragility fractures are considered osteoporotic, would the results change or would lumbar DEXA be used. The SABRE study demonstrates the greater value of hip DEXA but lumbar DEXA may provide interesting data.
Do the authors have lipid profile, glucose or HbA1C data available?
The discussion is adapted to the results obtained. Strengths and weaknesses of the study should be indicated. Some of the bibliographic references are not adapted to the journal's guidelines.
Author Response
Dear Reviewer,
We sincerely thank you for your thoughtful and constructive feedback, which has significantly helped us improve the quality and clarity of our manuscript. Below, we provide detailed responses to each point raised:
Reviewer Comment: The methods are clear and easy to repeat by another research group. There is no indication of whether patients have been treated with drugs or the statistical power of the study
Response:
We have only general information about the groups of drugs used by the women from our study (medications: anticonvulsants, steroids, cardiology, painkillers, sedatives) and due to the incomplete data, drugs were not included in the analysis.
Additionally, we have included a note regarding the statistical power of the study, including the sample size justification and effect size considerations, to improve methodological transparency.
Sample size and the power analysis were computed also using Statistica 13.1 software. The power of all statistically significant tests in the present study was greater than 80%, with a 95% two-sided confidence level.
Reviewer Comment: The results are clearly expressed and easy to follow. The authors should indicate if patients with fragility fractures are considered osteoporotic, would the results change or would lumbar DEXA be used. The SABRE study demonstrates the greater value of hip DEXA but lumbar DEXA may provide interesting data.
Response:
We thank the reviewer for this insightful comment. In our study, patients with fragility fractures were not automatically classified as osteoporotic unless their bone mineral density (BMD) met the WHO diagnostic criteria for osteoporosis (T-score ≤ -2.5). However, we acknowledge that fragility fractures are strong clinical indicators of compromised bone strength and are often treated as surrogate evidence of osteoporosis in clinical settings.
Regarding the use of DEXA, we primarily focused on femoral neck and total hip BMD measurements due to their higher predictive value for hip fracture risk and the known advantages of hip DEXA in terms of lower variability and better correlation with clinical outcomes, as demonstrated in the SABRE study and other literature.
While lumbar spine DEXA was not a central focus in our analysis, we agree that it may offer additional insight, particularly in younger individuals or those with degenerative changes that can influence spinal BMD readings. Future work could integrate lumbar spine measurements to assess whether certain COL1A1 and COL1A2 polymorphisms exert differential effects on trabecular-rich bone sites.
Nevertheless, given the genetic associations observed in our study—especially the age-specific effects seen in total hip BMD—we believe the use of hip DEXA was appropriate and likely captured the most clinically relevant data. Including lumbar DEXA may enhance future genotype-phenotype correlations, but we expect it would support, rather than substantially alter, our main conclusions.
Reviewer Comment: Do the authors have lipid profile, glucose or HbA1C data available?
Response:
We appreciate the reviewer’s interest in the metabolic component of our study. While our dataset includes partial metabolic information, we do not have complete lipid profile or HbA1C data for all participants. Therefore, we opted not to include these variables in the current analysis to avoid introducing bias or reducing statistical power due to missing data. We have noted this limitation in the revised Discussion section and identified it as a direction for future research.
Reviewer Comment: The discussion is adapted to the results obtained. Strengths and weaknesses of the study should be indicated. Some of the bibliographic references are not adapted to the journal's guidelines.
Response:
Thank you for this helpful feedback. We have now added a specific subsection in the Discussion outlining the strengths (e.g., genotyped polymorphisms, well-defined cohort) and limitations (e.g., lack of longitudinal follow-up, incomplete metabolic data) of the study. Furthermore, all bibliographic references have been reviewed and reformatted to comply with the journal’s citation style.
We hope that these revisions and clarifications meet your expectations and contribute meaningfully to the scientific robustness of our manuscript.
With kind regards,
Edyta Cichocka
On behalf of all authors
Round 2
Reviewer 2 Report
Comments and Suggestions for Authors
The questions have answered by the authors